# “Stigma and HIV Are Like Brother and Sister!”: The Experience of African-Born Persons Living with HIV in the US

**DOI:** 10.3390/pharmacy8020092

**Published:** 2020-05-30

**Authors:** Alina Cernasev, William L. Larson, Cynthia Peden-McAlpine, Todd Rockwood, Paul L. Ranelli, Olihe Okoro, Jon C. Schommer

**Affiliations:** 1College of Pharmacy, The University of Tennessee Health Science Center, 301 S Perimeter Park Drive, Suite 220, Nashville, TN 37211, USA; 2Allina Health Uptown Clinic, 1221 West Lake St., Suite 201, Minneapolis, MN 55455, USA; William.L.Larson@allina.com; 3School of Nursing, University of Minnesota, 308 Harvard Street SE, Minneapolis, MN 55455, USA; peden001@umn.edu; 4School of Public Health, University of Minnesota, 420 Delaware Street SE, Minneapolis, MN 55455, USA; rockw001@umn.edu; 5College of Pharmacy, University of Minnesota, 232 Life Science Duluth, 111 Kirby Drive, Minneapolis, MN 55812, USA; pranelli@d.umn.edu (P.L.R.); ookoro@d.umn.edu (O.O.); 6College of Pharmacy, University of Minnesota, 308 Harvard Street SE, Minneapolis, MN 55455, USA; schom010@umn.edu

**Keywords:** stigma, persons living with HIV, African-born

## Abstract

Minnesota has seen an increase in the number of immigrants from Africa, notably in the mid-1990s, making up around 2% of Minnesota’s total population. This population also faces many impediments that cause important difficulties not only for HIV prevention but also for treatment and care options. The objectives of this study were to capture the experiences of Persons Living with HIV (PLWH) in Minnesota (US) and to elicit their stories about their diagnosis news and what management strategies they use for coping with the stigma associated with the disease. Participants were recruited via fliers in pharmacies, clinics, and HIV service centers located in Minnesota. Recruitment continued until thematic saturation was obtained. Fourteen subjects participated in audio-recorded, semi-structured interviews that were transcribed verbatim into written text. The transcriptions were analyzed using Thematic Analysis. Three themes emerged from the data. Theme 1: Cruel News: “HIV-Oooooo! I wish I was dead”, Theme 2: This is My Secret! and Theme 3: “Stigma and HIV are brother and sister”. The results demonstrate that stigma is an ever-present problem in African-born PLWH living in the US. Participants perceived the stigma associated with HIV status to affect their lives and culture at individual, familial, and societal levels.

## 1. Introduction

In the early 1980s, the landscape of infectious diseases changed dramatically due to the identification of a mysterious disease later termed human immunodeficiency virus (HIV). Initially, scientists around the globe were intrigued and challenged by this new disease that did not appear to resemble the characteristics of any other [1].

The scientific world was confronted by numerous cases reported globally. The World Health Organization (WHO) estimated that around 36.9 million people were living with HIV globally in 2017 [2]. Africa has the highest prevalence of HIV, where approximatively 25.7 million people were infected in 2018 [2]. Furthermore, out of 1.8 million new global infections, 1.2 million occurred in Africa in 2017 [2].

Globally, most HIV infections occur through heterosexual unprotected sexual contact with an HIV-positive person [3]. Infected females could transmit the virus to a newborn during pregnancy, delivery, and breastfeeding [4]. Sharing contaminated needles for injection represents another risk factor for the transmission of the virus. The virus can also spread among men who have sex with men (MSM) through unprotected sexual contact with an infected partner [5].

Recent studies showed that sustained adherence to antiretroviral medications in HIV-positive subjects and achieving viral load suppression decreases the chances of virus transmission over time and reduces the risk of infection and that effective antiretroviral (ARV) therapy results into no new HIV-1 cases [6]. For example, an observational study showed that there were no new cases of HIV transmission in HIV-serodifferent and MSM couples and without the usage of condoms [7]. A follow-up study (PARTNER2) of serodiscordant homosexual couples, which tested the viral load and HIV status of the appropriate partner, concluded there was no transmission per 100 couple-years of follow-up when the HIV-positive partner’s viral load was less than 200 copies/mL [8].

Clearly, an undetectable HIV viral load translates into HIV becoming untransmittable [9]. In October 2017, the CDC released a letter in which it endorsed a public statement, “U = U” or “Undetectable Equals Untransmittable” [9]. Furthermore, in October 2017, the Minnesota Department of Health also launched this “U = U” campaign. The main goal of this “U = U” campaign was to inform the public about the importance of taking ARV medications [10]. Ultimately, adherence to ARV medication also decreases the risk of drug resistance, improves quality of life, and ultimately increases life expectancy [11,12].

Furthermore, the “U = U” movement is aimed at dispelling stigma at the personal and societal levels. Thus, “U = U” provides additional incentives for PLWH to adhere to their antiretroviral regimen in order to achieve an undetectable viral load.

A vast majority of PLWH face many tribulations related to the stigma of this disease. Since the rise of this epidemic within the past few decades, society has recognized the significance of HIV; however, the stigma still remains dominant and is difficult to eliminate. Although stigma has not been solely associated with HIV for the past three decades, numerous studies have been conducted globally. Other disease states, including mental health and opioid use disorders, have been confronted with stigma. For example, researchers have shown that the stigma associated with mental illness represents an obstacle for patients to seeking help and treatment for the medical condition [13].

The origins of the stigma associated with HIV are multi-faceted, which have been attributed to a combination of factors including stricter religious beliefs, homophobia, and lack of understanding about the contamination process [14]. Furthermore, previous studies have revealed misconceptions, such as women who were HIV positive being labeled as having a promiscuous life [15,16].

The literature review analyzing the stigma within this disease population, identifies “tackling stigma and discrimination” as one of the five major factors necessary for success. Stigma impedes the individuals’ mental and personal growth, leaving them feeling lost and forsaken in the midst of their HIV status. PLWH should not have to go through the trouble stigma causes alone. In order for these individuals to live as positively and as normally as possible, there must be an intervention to remove the discrimination.

In the last decade, Minnesota has seen an increase in the number of HIV cases in African immigrants, representing 2% of the total Minnesotan population. This population faces impediments including a language barrier, a poor understanding of how to navigate the US healthcare system, and difficulties with preventing and treating HIV.

A literature review revealed three quantitative studies conducted in the US with PLWH who were born in Africa. These three studies were retrospective chart reviews of HIV-positive African-born patients who received antiretroviral (ARV) therapy in the US [17,18,19]. Akinsete et al. investigated the HIV-1 genotyping and subtyping of infection in HIV-positive African-born patients. This Minnesota-conducted study provides information on the demographics of the population of interest. This literature review shows a gap in findings based on qualitative studies conducted in the US. These studies did not discuss the psychosocial effects of living with HIV in the US. Additionally, the literature search showed a limited number of qualitative studies conducted globally. For example, three studies conducted in London, United Kingdom presented the social–economic factors faced by this population in a country with different healthcare settings compared to the US [20,21,22]. The current study sought to bring more value to the literature by exploring the psychosocial effects of HIV in African-born PLWH in the US.

The limited literature on this understudied population encourages an in-depth examination of the experiences of African-born persons living with HIV in the US. The objectives of this study were to capture the experiences of PLWH in the US regarding their HIV diagnosis, management, and coping strategies.

## 2. Material and Methods

### 2.1. Study Design

A Narrative Inquiry method was used for this study. Narrative Inquiry is an inductive research method that elicits a participant’s stories about an event in his or her life and does not rely on a theoretical framework for eliciting the story. This method was selected because it is consistent with the specific objectives of this study. The study was approved (STUDY00001597) by the University of Minnesota Institutional Review Board (IRB).

### 2.2. Participant Recruitment and Data Collection

Participants for this study were recruited via fliers placed in pharmacies, clinics, and HIV service organizations in the urban areas of Twin-Cities, Minnesota between 2017 and 2018. Participants were recruited via fliers by the Principal Investigator if they self-identified as receiving HIV treatment and were Minnesotan residents. Participants were eligible if they were 18 years or older, spoke English, and were born in Africa. Prior to each interview, the anonymity of the participants was ensured by obtaining verbal informed consent, which was appropriate for a high-risk and vulnerable population [23]. To further prevent a link between the audio recordings and the participants, no demographic or identifiable information was collected during the interview. The Principal Investigator provided the consent form to each participant, and questions were asked as to if they understood what was conveyed in the oral consent form. The consent form was prepared in accordance with the IRB’s rules and regulations from the University of Minnesota.

All the interviews were conducted using a semi-structured guide and were audio recorded. The interviews were conducted in safe, private spaces per convenience and the preference of each participant. The duration of the interviews spanned from half an hour to two hours. The objective of the interviews was to explore and gather participants’ experiences of living with HIV and its associated stigma. One of the central questions asked to the participants was “Can you talk to me about your culture and if it is it OK to talk about HIV?”. Additional clarifying questions were asked to obtain more comprehensive information on their daily experience of living with HIV.

All audiotaped interviews were transcribed verbatim by a HIPAA-compliant transcription company. The transcription company was used to avoid any bias that could be introduced during transcription. During the transcription process, a supervisor checked the audio recordings in comparison to the transcript for accuracy. The Principal Investigator also listened to two audio recordings in order to ensure the accuracy of the data. This method served as a form of verification before data analysis was initiated.

### 2.3. Data Analysis and Establishing Rigor

Thematic analysis was used to analyze the qualitative interview data. Braun and Clarke recommend using an inductive and reflexive process for conducting a thorough and transparent data analysis. The data analysis followed the steps suggested by Braun and Clarke. For example, after the familiarization with the data, the transcripts were inductively coded and codes were generated. Similar codes were grouped into categories. All the categories were clustered and analyzed to uncover the major themes of the experiences of African-born PLWH living in the US and the stigma associated. Dedoose, a qualitative analysis software package, was used for generating initial codes and developing and reviewing themes. The recruitment of the participants continued until thematic saturation was achieved, at which point no new themes emerged with subsequent interviews [24].

To ensure and establish the rigor of the data, Lincoln and Guba’s criteria were implemented throughout the study in terms of the data collection, data analysis, and findings. For example, the codes were verified by two researchers to ensure that the identified codes and themes were appropriate. Additionally, the team met to validate the appropriateness of the extracted themes. For transparency, the study results used vivid and rich example quotations from the raw data that reflect the lives of African-born PLWH [25].

## 3. Results

The Thematic Analysis revealed the experiences of African-born PLWH in the US and the stigma associated with their diagnosis. The Thematic Analysis also illuminates the context of their experience of being diagnosed with HIV and transitioning to an unfamiliar US healthcare system. All the participants were born in Africa and represented seven countries. Eight out of fourteen participants were female.

### 3.1. Theme 1: Cruel News: “HIV-Oooooo! I Wish I Was Dead”

This theme illustrates the participants’ emotional suffering at the moment of receiving the devastating news of being HIV positive. Each of the participants had a different story. However, some of the common characteristics of their stories were the emotional states brought by the diagnosis news, such as the immediate despair, misery, suicidal thoughts, and denial of the HIV status.

“*Oooooo! I wish I was dead*”. This quote was selected to depict the first theme because it was representative of both the characteristics of the devastating diagnosis news and the denial aspect stated by the participants. All the participants stated that they were not emotionally prepared to receive the shocking news. Some of the participants described the distressing news emotionally by using words such as “depression”, “crying”, or “suicide”. Furthermore, a few participants reported their wish of not being around following the diagnosis.

Some participants were not ready to share their stories about when they were diagnosed with HIV. One of the main reasons for not sharing their stories was the emotional state it triggered when remembering those difficult moments of their lives. Even for those who were willing to share their stories, a lot of emotion could be heard in their voices. Tears would come to their eyes. Some began to speak at a hurried pace, while others needed a few moments to pull themselves together after disclosing those “wounds”. For example, one participant briefly mentioned the devastating moment of hearing about the diagnosis while in prison and moments before giving birth to her baby. Another participant could not talk about her rape, and another participant was not ready to share additional information from when she was diagnosed before giving birth.

The agony of receiving the disturbing news of being HIV positive impacted all the participants emotionally; however, their reactions were different and are presented in the following examples. One participant (P8) described the emotional disbelief when he received the diagnosis and highlights how the cruel news affected him by showing signs of depression:
“Oooooo! I wish I was dead. When I heard it, HIV, I know before I got friends, get friends and start some of them they are-you know, HIV they had it, they live, but I would never have thought HIV, you know, I can get it!… I wished I was dead when I-I was devastated when I found out you know, I was, you know HIV positive.”.

Another participant (P12) outlines the “depression symptoms” that he experienced after receiving the diagnosis of being HIV positive:
“When I got diagnosed first, I was depressed and discouraged. I felt that my whole life had crushed down, but when I start seeing other people with HIV… when I started going to… and see how people are living a vibrant life.”.

Frequently, the concept of “being dead” rather than having HIV was mentioned by the participants to varying extents:
“I used to cry all day long… I said, ‘I wish I would just die if that was what I had. If that was caused it, yea…’ She {the nurse} told me you got to know the right place because now there is a treatment for anything… So don’t get discouraged to that.”(P3)

This impactful moment was described by the following participant with a single word, “suicide”. Note how he described that precise moment of receiving his HIV status. In the participant’s statement below, there is a strong recollection of the moment that had changed his life forever. Note that this participant avoids using the word HIV. He refers to HIV as “this”. He says:
“February 2001, I have been told that I have this. It was a Friday around 4 o’clock PM. Uh because myself I don’t know too much about the disease uh, like I say for us when you have HIV it’s the death now or tomorrow. Uh, I was about to take my own life. That same Friday… I came to the [NAME] clinic… They gave me the information I came in the bus from North Minneapolis. I was living with a friend of mine. They gave me the information and they let me go by myself… and in my way back home in the bus I was telling myself ‘This is it, I need-I cannot handle this.’”(P14)

### 3.2. Theme 2: “This Is My Secret”

After the participants received the devastating news, their HIV diagnosis reportedly became their “secret”. In describing their stories, the participants pointed out the emotional difficulty they faced when thinking about how to reveal their “secret” to loved ones, including parents or children. Each of the participants had a different story and reason for keeping the diagnosis a secret. However, similar to each of their stories were the emotions brought by the news of diagnosis, such as immediate despair, misery, suicidal thoughts, and denial. Furthermore, the participants described their fear of the consequences they would face if they disclosed their status to immediate family members:
“And uh, when the people know that you are HIV, people starting to separate you, like, yea like outcast.”(P7)

Participant 10 could not find the strength to share her status with her mother. Using the negative form of the verb, as she did multiple times, reinforces the secrecy of her HIV status:
“I’ve-up to now I’ve not talked-I’ve not told my mom. I’ve not talked in my mom presence. I didn’t, I didn’t, I didn’t!”.

Several participants stated that they were not able to share their diagnosis with their blood relatives due to preconceived notions concerning the HIV diagnosis. According to most participants’ stories, HIV is still associated with a death sentence in their country of origin. The fear of being isolated by society was another contributing factor to keep their “secret”:
“Even your own family is afraid of you because they don’t want to die. They know it’s something that kills, it has no cure. So, if they don’t know that just by you sitting by me will not make you to have it, they get afraid, they don’t want to die.”(P3)

The fear of disclosing the secret was echoed by P10, who uses the word ‘’outcast’’ to describe her emotions:
“… So, anyone who had HIV was considered an outcast. As if no one wanted to come to be with you to even if they knew, they wouldn’t even come to my house, my relatives. If they knew, my friends would not even come. You know, sharing cups-even in the house like usual.”.

### 3.3. Theme 3: “Stigma and HIV Are Brother and Sister”

This theme emerged without prompting the participants to discuss it. The quotation “Stigma and HIV are brother and sister” was the most significant, concise, and explanatory phrase that captured the stigmatizing situations faced by participants. This quotation also presents the magnitude of the defaming impact of HIV status on the participant. Participants used different terminology such as “isolation”, “separation”, and “abandonment” to describe “stigma” during the interviews.

The participants attributed the stigma effect to the existence of misconceptions about HIV transmission in their country of origin. Firstly, at the societal level, participants linked the stigma of being HIV positive with their country’s cultural norms and values and the lack of public discussion about sexual reproduction. Secondly, at the individual and societal level, participants presented the link between persistent misconceptions in their culture and HIV transmission. Thirdly, at the familial and community level, participants connected the misconceptions of HIV transmission and the lack of family support to the fear of disclosing their HIV status. The sub-themes that emerged from their narratives were:The link between stigma and the cultural values of their country of originThe landscape of societal misconceptions about HIV transmission in their country of origin

#### 3.3.1. Sub-Theme: Link between Stigma and the Cultural Values of Their Country of Origin

The first sub-theme highlights the participants’ perspectives of the role played by the cultural norms and values of their country of origin in the association of stigma with HIV status. When asked if it is acceptable to talk about HIV in their country of origin, all the participants had a negative response.

For instance, Participant 12 clarified the reason why it is impossible to openly discuss the issues related to sex:
“… in my culture sex is not public discussion. People cannot sit down, you know, and open forum and discuss about sex, so sex is like a private …”.

Some participants illustrated disgrace and isolation as the effects of stigma perceived at the societal level. Participant 6 used a powerful, defaming, and insulting comparison between HIV positive people and “animals” to depict the stigma felt by him and his peers:
“Oh, before they used to take you as if a wild. Wild beast animals, before. Everywhere in Africa even. But not now, now-the activism work is done, nicely, everywhere. The graph is- not increasing, decreasing. The segregation is decreasing. You can sit, you can enjoy, you can eat. You can drink with them, no problem…”(P6)

Furthermore, one participant pointed out a different aspect that connects taking ARV medications and HIV stigma. She stated that some HIV-positive people might not take their ARV medications due to misbeliefs that persist in their country of origin:
“…and there are so many Africa-like, some of them do not take their medications. They will say, ‘Hm, they told you you have HIV, I don’t think you have it, but the moment you start taking HIV pills, you gonna have it [as in: paranoia that the medication carries the infection].’ That’s why they don’t take their medications.”(P3)

#### 3.3.2. Sub-Theme: The Landscape of Societal Misconceptions about HIV Transmission in Their Country of Origin

This sub-theme presents participants’ opinions about the presence of societal misconceptions of HIV transmission in their country of origin. Numerous participants shared their stigma experiences with social misconceptions of HIV transmission including transmission via shaking hands, sitting next to someone, or sharing the same dishes. This sub-theme also illustrates the fact that the social misconceptions in these countries of participants’ origin affected their lives at the individual and interactional levels. A few of the participants highlighted that misconceptions persist in the US in some of the African-born individuals.

One fundamental societal misconception in participants’ countries of origin was labeling female PLWH as sex workers. Several of the female participants expressed that their community would slander the women who were HIV positive and would portray them as sex workers. For instance, the use of the word “prostitute” in this extract highlights a situation where the female participant is stigmatized due to the HIV diagnosis:
“No, they talk to you, but you just a subject of gossip. I mean… I don’t know they feel that once you have HIV, you have been a prostitute, have been sleeping all around, and that’s how you got it. That is one of the main point of gossip—”(P3)
“Actually I’ve-up to now I’ve not talked-I’ve not told my mom. I’ve not talked in my mom presence. You know, OK, as I said back then, HIV was labeled like. If you’re HIV you’re a prostitute. If you’re HIV you’re-I don’t know they just, they associated HIV with prostitution, so anyone who had HIV was considered an outcast.”(P10)

Another societal misconception that emerged from the interviews was that the disease could be transmitted via a meal. It was highlighted by different participants that there were situations when the HIV-positive person would receive a separate dish to ensure the disease would not be spread to friends or family members. Another societal misconception that emerged was that shaking hands with an HIV-positive person would lead to the transmission of the virus.

P3 explained that these societal misconceptions vanished from her mind once she received appropriate educational information about the transfer of the virus. However, these societal misconceptions about HIV transmission continue to exist in her community within the US. Because these misconceptions persist in her society, the participant would not disclose her status to her church friends and acquaintances. Revealing her HIV status to the church community would result in distancing her from that community.
“We thought it was like contagious just by shaking your hand or sitting with you. We didn’t know it was contagious in other ways. Like seizures, we will have seizures, we are afraid. We said, ‘If this touches you, you’re gonna have seizures.’”(P3)

P14 reinforces the above societal misconceptions that HIV is transmitted by touching:
“So they are afraid even to touch you. They are afraid. Because they need more explanation about the whole thing.”(P14)

## 4. Discussion

This narrative study explored African-born PLWH experiences from the moment of diagnosis and the stigma associated with the diagnosis in the home country and the US. Three themes emerged during the analysis: *Cruel News: “HIV-Oooooo! I wish I was dead**”*, *This is My Secret!*, and “*Stigma and HIV are brother and sister*”. The findings illustrate that all the narratives presented in this study are distinguished by secrecy regardless of when the participants were diagnosed with the disease. The secrecy of diagnosis was a common theme throughout all interviews, where the participants would refrain from using the word “HIV” and refer to it as “this”, “it”, or “the disease”.

Being in denial of one’s HIV status is a complex phenomenon that has been extensively researched [26,27]. One study conducted in Tanzania explored the association between stigma, revealing HIV status, coping mechanisms that included denial or acceptance, and adherence to ARV medications [28]. The study showed that denial is a defense mechanism that has been attributed to perceived stigma [28]. The denial aspect has been negatively associated with lower adherence to ARV therapy [29]. However, in our present study, none of the participants reported decreased adherence to their ARV regimen due to denial.

The denial aspect of the diagnosis in our findings resonates with the Ross theoretical framework. During this stage, the participants cannot accept the cruel news of diagnosis. The following sentence is representative of most participants regardless of their diagnosis: “No, not me, it cannot be true.” [30]. A previous study applied the E.K. Ross framework to PLWH. The denial stage of the framework showed that participants had different manifestations during this stage. For example, some were unwilling to be tested for the disease, despite the fact that they were presenting the characteristic symptoms, while others were not ready to start the ARV treatment. Our findings corroborated the previous study and showed that the theme of patient denial is ubiquitous.

The stressful aspect of revealing a HIV-positive status to others has received a great amount of attention in research and has been associated with the fear of discrimination. External judgement and discrimination perceptions were expressed by this study’s participants and are mainly the result of the stigma associated with HIV. Each participant in this study tended to hide their diagnosis from others as a result of shame. These findings are consistent with the literature that showed many PLWH are not disclosing their status due to the fear of potential discrimination [31].

A wealth of research has been conducted on the stigma associated with HIV [32,33]. There are limited definitions of the stigma; however, the literature uses terms such as “discrimination”, “shame”, or “fear” [34]. Stigma terminology is ambiguous and mostly focuses on the emotions and situations experienced by PLWH [34]. Some of the participants in the present study used different terminology to describe stigma.

Stigma experiences have been linked to denial by relatives or friends, shame, and solitude. The participants in this study inferred stigma as shame and isolation. Link and Phelan proposed a conceptual framework for stigma. According to the proposed framework, stigma has four components and it occurs when these components converge. Per the framework, the four components are “labelling”, “negative attributes”, “separation”, and “discrimination”. The findings of this study resonate with the Link and Phelan conceptual framework. For example, this study showed that the participants were afraid of the consequences of being “labelled” as disease carriers. Once someone knew their HIV-positive status, a cascade of events would unravel that would result in “discrimination”, the last component of the framework. Furthermore, this study uncovered a unique finding, which is the persistence of stigma in participants’ communities. Even though they live in the US where they feel less stigmatized by Americans, they still feel stigmatized in their community where the population is predominantly of African heritage.

Lastly, as described by the framework, the status loss is the “immediate consequence of successful negative labelling and stereotyping resulting in a general downward placement of a person in a status hierarchy”. A number of participants highlighted the fact that many people would be afraid to be associated with PLWH. For Participant 11, the status loss was expressed by use of the word “ostracized”.

In many of the participants’ narratives, stigma at the societal level was attributed to misconceptions about transmission such as through shaking hands, sharing food utensils, or sitting next to PLWH. Despite the fact the participants came from different African countries, some of the misconceptions were common across these countries. Additionally, in a few narratives, female participants challenged the misconception that associates the diagnosis of HIV with immorality and promiscuity.

Even though a wealth of research has been conducted globally on the stigma associated with PLWH, there are limited studies conducted across different countries in Africa. One study conducted in six different African countries cross-culturally compared the content of stigmatizing descriptions with the salient goal of decreasing the stigma [35].

## 5. Strengths and Limitations

To our knowledge, this is the first study to take a narrative approach in exploring African-born PLWH experiences in the US and the stigma associated with HIV. Every interview contained rich data that enabled the extraction of the themes and obtaining of the thematic saturation of the data. All the interviews were anonymous and specific. Demographic data about the participants’ levels of education were not collected. Most of the participants mentioned during the interview that they have a higher level of education in the country of their origin. Subsequently, most of them spoke English as their primary language. Furthermore, because of the anonymity of the data, the diagnosis of the participants was not confirmed by medical charts and the study relied on self-reporting.

## 6. Implications for Practice

The results of this study have implications for healthcare practice. Pharmacists and healthcare providers need to understand the stigma many PLWH face and address these patients with compassion. Whenever pharmacists and other healthcare providers encounter PLWH who were born in Africa, they should have open discussions with them to better address the stigma associated with the diagnosis. During counseling, pharmacists can assure PLWH of African heritage that their diagnosis is kept confidential and will not be shared with other people, including the family. Counseling could be the premise of building trusting relationships and providing education on HIV to reinforce the “U = U” concept that helps PLWH overcome stigma, particularly perceived stigma. Given that participants frequently expressed maintaining secrecy regarding their diagnosis, when pharmacists and other healthcare professionals understand the personal challenges PLWH face, they can be better allies to these patients, who likely have no one else to turn to.

Furthermore, the study findings suggest that to better meet the needs of African-born PLWH, community awareness is necessary to better understand the stigma issues and transmission of HIV.

The knowledge learned from this study suggests there are many ways to decrease the stigma associated with HIV diagnosis. Community leaders, pharmacists, and other health providers could learn how to be more culturally sensitive when interacting with PLWH of African heritage.

## 7. Conclusions

The study results show that stigma is an ever-present problem for the African-born PLWH. Even though the participants live in the US, they perceived the stigma associated with HIV status persisting in their culture at individual, familial, and societal levels.

The present study offers key information regarding why participants keep their diagnosis as a “secret.” The fear of not disclosing the “secret” with blood relatives was attributed to the external stigma brought upon themselves and how others would perceive them.

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
