# Peer review of "“Stigma and HIV Are Like Brother and Sister!”: The Experience of African-Born Persons Living with HIV in the US"

_pharmacy, 2020, doi:10.3390/pharmacy8020092_

Round 1
Reviewer 1 Report
Cernasev and colleagues conducted a narrative study that explored the experiences of African-born persons Living with HIV (PLWH) in Minnesota and the stigma associated with the diagnosis in the home country and the US. The study enroled 14 participants in total who were participated in audio recorded, semi-structured interviews that were transcribed verbatim into written text. The transcriptions were analyzed using Thematic Analysis and the authors reported that three themes emerged during the analysis: Cruel News: “HIV-Oooooo! I wish I was dead,” This is My Secret! and “Stigma and HIV are brother and sister.” The peer-reviewed manuscript is poorly written, for instance, introduction is extensive, information provided are out of age and cited references are too old. The study in total does not add any significant information to the already published relevant literature and is not suitable for Pharmacy journal, since it does not falls into any of the subjects’ areas that are of interest for this journal.
Author Response
Reviewer #1
Cernasev and colleagues conducted a narrative study that explored the experiences of African-born persons Living with HIV (PLWH) in Minnesota and the stigma associated with the diagnosis in the home country and the US. The study enroled 14 participants in total who were participated in audio recorded, semi-structured interviews that were transcribed verbatim into written text. The transcriptions were analyzed using Thematic Analysis and the authors reported that three themes emerged during the analysis: Cruel News: “HIV-Oooooo! I wish I was dead,” This is My Secret! and “Stigma and HIV are brother and sister.” The peer-reviewed manuscript is poorly written, for instance, introduction is extensive, information provided are out of age and cited references are too old. The study in total does not add any significant information to the already published relevant literature and is not suitable for Pharmacy journal, since it does not falls into any of the subjects’ areas that are of interest for this journal.
Response:
Thank you for your time in reviewing our manuscript. In light of the other reviewers' comments, we decided to keep the introduction and added a paragraph. Furthermore, this manuscript is being submitted for a special issue on Medication Experiences and this special issue invited papers of many different types including commentaries, case studies, and research conducted in different areas of the public health arena.
It was pointed out to us that the references were old; however, there are no new references on how the disease is spread since HIV was identified over three decades ago.
The pharmacists reading this manuscript might learn from this study. For example, there is a sizeable African-born population in Minnesota and other states, where it is crucial for the pharmacists (especially the community pharmacists) to know the implications when interacting with this population. Furthermore, these participants keep their diagnosis secret, and as seen in the manuscript, would not share it with loved ones and other healthcare professionals. Therefore, with this knowledge in mind, pharmacists could learn how to interact with this understudied population. We tried to explain the reason why the participants might not want to disclose their diagnosis due to confidentiality reasons. Additionally, the HIPAA law has been enforced in the U.S. for over two decades. Breaching the HIPAA law would have severe consequences for the pharmacist and the pharmacy.
Reviewer 2 Report
- Social and Emotional Stigma in PLWH have been one of the major concerns for the medical society.
- This article by Cernasev et al. shed some light on the different psychosocial components that PLWH have to go through based on three themes such as Cruel News, This is my secret and Stigma-HIV.
- This reviewer does not have major question over the article but has some basic concerns in general as follows.
- This article mainly deals with a social/public aspect and mental/emotional concerns associated within it, so I wonder if it is suitable for this journal as it does not pertain to pharmacy or any pharmaceutical aspects of the disease? It would be more suitable for any epidemiology related journal.
- I was not sure how many subjects were recruited to the study?
- Were there any exclusion criteria as well and control measures?
- Was any medium or source was provided to these subjects for overcoming the society stigma fear?
- A thorough read proof of the article is needed.
Author Response
Reviewer #2
- Social and Emotional Stigma in PLWH have been one of the major concerns for the medical society.
- This article by Cernasev et al. shed some light on the different psychosocial components that PLWH have to go through based on three themes such as Cruel News, This is my secret and Stigma-HIV.
- This reviewer does not have major question over the article but has some basic concerns in general as follows.
- This article mainly deals with a social/public aspect and mental/emotional concerns associated within it, so I wonder if it is suitable for this journal as it does not pertain to pharmacy or any pharmaceutical aspects of the disease? It would be more suitable for any epidemiology related journal.
- I was not sure how many subjects were recruited to the study?
- Were there any exclusion criteria as well and control measures?
- Was any medium or source was provided to these subjects for overcoming the society stigma fear?
- A thorough read proof of the article is needed.
Point 1: Thank you for this suggestion regarding a different journal rather than the pharmacy profession. However, this manuscript is being submitted for a special issue on Medication Experiences and this special issue invited papers of many different types including commentaries, case studies, and research conducted in different areas of the public health arena.
Consequently, this manuscript could be read through the lens of a retail pharmacist who could interact with a patient who is originally from Africa and does not want to share her/his diagnosis. In our opinion, the pharmacist could learn from this study how to interact and provide care for these patients. Furthermore, most of the participants are not aware of the fact that the US pharmacists cannot breach the HIPAA rules and share their diagnosis with anyone. Breaching the HIPAA law would have severe consequences for the pharmacist and the pharmacy.
Point 2: The study recruited the participants until saturation was obtained. Once the saturation of the data occurred the study stopped recruiting the participants. Therefore, the answer is 14 participants.
Point 3: The main exclusion criteria were: participants who did not speak English, were not born in Africa, and were HIV positive, but did not take antiretroviral therapy at the time of the interview.
Point 4: Thank you for the suggestion; however, the study could not provide any medium or any other source for these participants to overcome the stigma fear. During the interviews, some participants mentioned that talking to the researcher had a healing effect on them.
Point 5: Thank you for this recommendation. We thoroughly revised the manuscript.
Reviewer 3 Report
This work adds significant knowledge on the topic of African immigrants health. However, it requires a couple of minor corrections:
- l34: The authors may change the word 'could' for 'can'
- l46: an introduction is needed on how important stigma is in relation to numerous diseases and its consequences
- l55: More details are needed on the increase. How much?
- Methods sections: A sentence is needed stating that demographic data was not collected
- Results: quotes from participants should be presented in italics
- The conclusions need to be more elaborate as it fails to include what is already known on the topic of HIV and stigma. The authors need to refer to specific features in different African cultures that may play a role
Author Response
Reviewer # 3
This work adds significant knowledge on the topic of African immigrants health. However, it requires a couple of minor corrections:
- l34: The authors may change the word 'could' for 'can'
- l46: an introduction is needed on how important stigma is in relation to numerous diseases and its consequences
- l55: More details are needed on the increase. How much?
- Methods sections: A sentence is needed stating that demographic data was not collected
- Results: quotes from participants should be presented in italics
- The conclusions need to be more elaborate as it fails to include what is already known on the topic of HIV and stigma. The authors need to refer to specific features in different African cultures that may play a role
Point 1: l34: Thank you for this suggestion. We changed it.
Point 2: l 46. Thank you for this recommendation. We added a paragraph about other disease states and how stigma associated with HIV is so important to this disease state.
Point 3: Methods section. Thank you for this recommendation. We amended the text to reflect that the demographic data was not collected.
Point 4: Results: Thank you for this recommendation. We amended the text and all the quotations are in italics.
Point 5: Conclusions: Thank you for this suggestion. We made the changes in the conclusion to reflect them.
Round 2
Reviewer 1 Report
Cernasev and colleagues in the revised manuscript have addressed/answered all the comments/suggestions pointed out by the other two reviewers.
Minor comment
In their response the authors state that “there are no new references on how the disease is spread since HIV was identified over three decades ago” and in the introduction there is a paragraph describing routes of HIV infections/transmissions (lines 42-46). Although these are correct in principle nowdays there is enough evidences supporting the U=U statement and the fact that effective ART gets to zero new HIV-1 infections (e.g N Engl J Med 2011; 365:493-505 JAMA. 2016 Jul 12;316(2):171-81, and Lancet. 2019 Jun 15;393(10189):2428-2438)
Author Response
In their response the authors state that “there are no new references on how the disease is spread since HIV was identified over three decades ago” and in the introduction there is a paragraph describing routes of HIV infections/transmissions (lines 42-46). Although these are correct in principle nowdays there is enough evidences supporting the U=U statement and the fact that effective ART gets to zero new HIV-1 infections (e.g N Engl J Med 2011; 365:493-505 JAMA. 2016 Jul 12;316(2):171-81, and Lancet. 2019 Jun 15;393(10189):2428-2438)
Point 1: Thank you for your recommendation and suggesting the articles. We amended the manuscript and wrote two paragraphs that described U=U and presented a summary of the suggested articles. Thank you once again for this recommendation that strengthens the manuscript.
Reviewer 3 Report
All of the comments have been addressed.
Author Response
All of the comments have been addressed.
Point 1: Thank you for your time to review our manuscript.